



# Aerosol single scattering albedo derived by merging OMI/POLDER satellite products and AERONET ground observations

Yueming Dong[1], Jing Li[1,2], Zhenyu Zhang[1], Chongzhao Zhang[1], and Qiurui Li[1]

[1]Laboratory for Climate and Ocean-Atmosphere Studies, Department of Atmospheric and Oceanic Sciences, School of Physics, Peking University, 100871, Beijing, China
[2]Collaborative Innovation Center on Forecast and Evaluation of Meteorological Disasters (CIC-FEMD), Nanjing University of Information Science & Technology, 210044, Nanjing, China

**Correspondence:** Jing Li (jing-li@pku.edu.cn)

**Abstract.**

Accurate global aerosol single scattering albedo (SSA) data is critical for assessing aerosol radiative effects and identifying aerosol composition. However, current satellite-based SSA retrievals are both limited and highly uncertain, whereas the more accurate ground-based observations lack global coverage. In this study, we employ an Ensemble Kalman Filter
(EnKF) data synergy technique to construct two monthly mean SSA datasets over land by synergizing OMI and POLDER with AERONET observations respectively, namely Merged-OMI and Merged-POLDER dataset. The background ensemble is constructed with 231/106 members using all monthly mean OMI/POLDER SSA available to represent the variability of SSA field. Then AERONET measurements are assimilated into each satellite dataset using the EnKF approach. The merged datasets show substantial improvements against the original products, with the correlation coefficient increased by up to 100%, and the
mean absolute bias (MAB) and root mean square error (RMSE) reduced by more than 30% compared with AERONET results. Cross validation using independent AERONET observations shows an average increase of 70% in correlation, 15% reduction in RMSE and 14% reduction in MAB for Merged-OMI dataset, and similar although weaker improvement for Merged-POLDER mainly due to the smaller sample size. This study confirms the effectiveness of the EnKF technique in extending the information obtained from ground stations to larger regions. The two merged datasets generated in this study can offer more accurate
SSA estimates for assessing aerosol radiative forcing and improving climate modeling, serving as an important resource for advancing global aerosol research.

## 1 Introduction

The scattering and absorption property of aerosols, denoted by the single scattering albedo (SSA) parameter, is critical for assessing aerosol radiative and climate effects (Hansen et al., 1997; Li et al., 2022; Ramanathan et al., 2001), as well as
characterizing aerosol type (Dubovik et al., 2002; Omar et al., 2005; Zhang and Li, 2019). It has been identified as a major source of uncertainty in quantifying aerosol climate effects, contributing to over 30% of the uncertainty in aerosol direct forcing in climate models (Loeb and Su, 2010; Zhang et al., 2022). Moreover, since most satellite-based aerosol optical depth (AOD) retrieval algorithms rely on assumed SSA values, inaccuracies in these assumptions can lead to significant errors in AOD



retrievals (Mielonen et al., 2011; Wu et al., 2016; Zhang et al., 2024). Therefore, accurate knowledge of global SSA is needed
to reduce uncertainties in global climate change assessments and improve satellite-based AOD retrievals (Kahn, 2012; Li et al., 2022; Thorsen et al., 2021).

Aerosol SSA observations can be derived from satellite- and ground-based remote sensing platforms. Ground-based remote sensing can provide SSA measurements at relatively high accuracy (Dubovik and King, 2000; Sinyuk et al., 2020). For example, SSA retrieved from the Aerosol Robotic Network (AERONET) ground-based sunphotometers can have uncertainties within ±0.03 when AOD exceeds 0.4 at 440 nm, and is typically used to validate satellite retrieval products (Dubovik and King, 2000). However, the spatial coverage of ground-based stations with consistent SSA measurements is still quite limited, whose SSA data is far from enough for large-scale global aerosol studies. On the other hand, satellite remote sensing is an indispensable tool in obtaining global SSA information for its extensive coverage. However, global satellite SSA products are largely limited and highly uncertain due to the great difficulties in its retrieval.

The reflectance at the top of atmosphere (TOA) observed by satellites comprises a mixture of signals from various sources, including aerosols, gas molecules, and surface. Most conventional sensors, such as Moderate Resolution Imaging Spectrora-diometer (MODIS), only measures single-view radiation intensity in the visible to near-infrared bands and have insufficient information to retrieval SSA (Dong et al., 2023; Levy et al., 2013). Even with more advanced satellite instruments, retrieving SSA remains challenging due to its high sensitivity to surface reflectance and aerosol vertical distribution (Li et al., 2022; Mahowald and Dufresne, 2004). Currently, global operational satellite SSA products are mainly derived from ultra-violet (UV) or Multi Angle Polarization (MAP) observations. UV sensors, such as Ozone Monitoring Instrument (OMI) and Tropospheric Ozone Monitoring Instrument (TROPOMI) can measure SSA according to the upwelling spectral dependence of Rayleigh scattering over sufficiently dark surfaces (Torres et al., 1998, 2020). OMI has been providing SSA retrievals using this method for over 20 years. However, the UV-based SSA products from OMI exhibit significant uncertainties (Jethva et al., 2014), pri-marily due to the high sensitivity of UV method to aerosol vertical distribution (Torres et al., 2007). The MAP configuration is widely regarded as the most promising satellite technique for retrieving SSA. The multi-angle view geometry can effectively separate aerosol and surface signals, while the polarization measurements provide high sensitivity to aerosol microphysical properties (Dubovik et al., 2019; Mishchenko et al., 2007). SSA retrievals from MAP instruments, such as the Polarization and Directionality of the Earth's Reflectance instrument (POLDER) and the Directional Polarimetric Camera (DPC), have been validated with moderate accuracy against the AERONET observations, although both of them exhibit obvious systematic bi-ases (Chen et al., 2020; Dong et al., 2024). Other approaches, including combining satellite and ground-based observations or integrating data from multiple satellite sensors, have also been explored but fail to offer global long-term SSA products due to limited general applicability (Devi and Satheesh, 2022; Dong et al., 2023; Lee et al., 2007). Overall, the existing satellite SSA products usually suffer from significant uncertainties and cannot meet the accuracy required for constraining aerosol radiative forcing on a global scale.

As such, a promising approach to obtain accurate global SSA observations is to combine the advantages of ground based and satellite measurements. Numerous studies made attempts to merge aerosol products from multi-sensors with ground-based observations, using various data-fusion approaches, such as the spatial statistical fusion method (Jinnagara Puttaswamy





et al., 2014; Nguyen et al., 2012), the universal kriging method (Chatterjee et al., 2010; Zhao et al., 2017), the Bayesian
maximum entropy method (Tang et al., 2016; Zhu et al., 2023), and the maximum likelihood estimate method (Xu et al., 2015),
etc.. In particular, Li et al. (2020) developed a data synergy method based on Ensemble Kalman Filter (EnKF) and applied
it to produce a merged global AOD dataset using multi-source satellite and ground-based AOD measurements. The EnKF
approach constructs an ensemble dataset with sufficient members to capture the variability of AOD at each location and its
spatial covariability with other locations, which enables to assign appropriate weights to ground-based observations in the data
synergy, resulting in substantial improvements of merged data compared to individual satellite data. Therefore, this method
has demonstrated strong efficacy and general applicability in integrating ground-based and satellite observations. Overall,
the merged datasets show enhanced accuracy compared to individual satellite datasets and offer wider spatial coverage than
ground-based datasets, significantly contributing to improved climate models and more accurate estimates of aerosol radiative
effects (Kahn et al., 2023). However, the majority of these work focused on merging AOD and fine mode AOD, whereas there
is a significant lack of research on multi-source data fusion for SSA.

In this study, we attempt to generate two merged global land SSA datasets by respectively combining OMI and POLDER
products with AERONET observations. Our data synergy method follows the EnKF approach developed by Li et al. (2020).
OMI and POLDER are separately merged with AERONET due to the large difference and bias between both the data quality
and record length of these two datasets. Section 2 describes the details of the methods and data used. Section 3 presents the
merged data and the validation of the results. A brief comparison between Merged-OMI and Merged-POLDER datasets is
discussed in Section 4, and the final section offers the conclusion with the acquisition of the merged datasets.

## 2  Data and method

### 2.1  Satellite SSA datasets

To construct an appropriate ensemble with relatively large spread in the data synergy, it is crucial to use satellite dataset
with long observation period and extensive global coverage. For this purpose, we utilize monthly mean SSA products derived
from OMI onboard Aura (Torres, 2015) and POLDER onboard Polarization and Anisotropy of Reflectances for Atmospheric
Sciences coupled with Observations from a Lidar (PARASOL) (Dubovik et al., 2011, 2014). These two sensors offer the longest
continuous global SSA observations available to date, and their SSA products have been validated with moderate accuracy.
Below is a detailed overview of these two sensors and related SSA products.

OMI onboard NASA's Aura satellite has been providing TOA reflectance measurements in three spectral bands ranging from
270 to 500 nm since 2004. A large swath of about 2600 km enables OMI to complete a global scan almost every day with a
nadir spatial resolution of $13 \times 24$ km. Here we use the monthly mean Level-2G global gridded SSA product OMAERUVG
at $0.25° \times 0.25°$ in 2004-2023, which is retrieved by the near-UV algorithm (Torres et al., 2007, 2013) (downloaded from
http://daac.gsfc.nasa.gov/). Jethva et al. (2014) assessed OMI OMAERUV SSA products and found 46% and 69% of OMI
SSA retrievals are within the error envelope (EE) of $\pm0.03$ and $\pm0.05$, respectively, compared to AERONET data. OMAERUV
algorithm performs better for carbonaceous and desert dust aerosols, with 52% and 77% samples falling into the EE of $\pm0.03$



and ±0.05. Based on the well-known sensitivity of Rayleigh scattering to aerosol absorption in the UV spectrum, the near-UV algorithm provides SSA retrievals at 354, 388, and 500 nm. For the data fusion, the OMI SSA retrievals are interpolated to the AERONET measured wavelength of 440 nm (Dong et al., 2023). We average the 0.25°×0.25° gridded data to 1°×1° as the

background field. The multiyear averaged monthly means are further removed to construct an OMI-based ensemble with 231 members using all monthly mean OMI SSA available. The standard deviation of all 0.25° grids within the larger 1° grid is used to represent the variability at each location in the OMI-based data fusion (as shown in Figure S1).

The third POLDER sensor onboard PARASOL provided the longest MAP observation records to date, operational from 2005 till 2013. POLDER features eight spectral channels ranging from 443 to 1020nm, including three polarimetric channels

centered at 490, 670, and 865 nm (Tanré et al., 2011). It can observe the target pixel from up to 16 viewing angles. MAP configuration of POLDER significantly enhances aerosol retrieval capabilities. Several algorithms have been developed for POLDER, yielding retrievals of multiple aerosol parameters with relatively high accuracy (Dubovik et al., 2019). Two representative algorithms are the Generalized Retrieval of Aerosol and Surface Properties (GRASP) (Dubovik et al., 2011, 2014) and the Remote sensing of Trace gas and Aerosol Products (RemoTAP) (Fu and Hasekamp, 2018; Hasekamp and Landgraf,

2007). Although the RemoTAP algorithm performs better in SSA retrieval than GRASP, only one year of POLDER/RemoTAP data is available with many missing values resulting from the strict quality control in the algorithm, which is insufficient to construct an ensemble with large spread. Here we use the monthly mean POLDER/GRASP High-Precision Level 3 SSA product (v1.2) at 0.1°×0.1° in 2005-2013 (Chen et al., 2020) (downloaded from https://www.grasp-open.com). Chen et al. (2020) evaluated the performance of POLDER/GRASP aerosol products, showing that SSA can achieve relatively high accuracy with

an $R$ value of 0.54 and a total bias of 0.03 (for GRASP/HP SSA at 670 nm). POLDER/GRASP SSA products at four bands (i.e., 443, 670, 865, 1020 nm), consistent with AERONET, are used in the data fusion after averaging to 1°×1° data. Similar to OMI, a POLDER-based ensemble is constructed with 106 members using all monthly mean POLDER SSA available. We also calculate the standard deviation of all 0.1° grids within each 1° grid as the representation error in the POLDER-based data fusion (as shown in Figure S2).

For clarity, the merged SSA based on the OMI OMAERUV product is referred to as the Merged-OMI SSA, while the merged SSA based on the POLDER GRASP High-Precision product is referred to as the Merged-POLDER SSA.

## 2.2   AERONET ground SSA measurements

The Aerosol Robotic Network (AERONET), operational since 1993, is the largest global network for ground-based aerosol monitoring (Holben et al., 1998). AERONET derives SSA at four discrete wavelength bands (i.e., 440, 670, 865, and 1020

nm) from both direct beam solar radiation and diffuse sky radiance measured by sunphotometers (Dubovik and King, 2000). AERONET SSA errors typically decrease as AOD increases, with errors generally within ±0.03 for AOD≥0.4 at 440 nm (Sinyuk et al., 2020).

Here we use Version 3 Level 2.0 monthly AERONET SSA products (cloud-screened and quality assured) (Sinyuk et al., 2020), ensuring that the daily SSA error is within ±0.03. A total of 525 sites provided Level 2.0 monthly SSA data during the

OMI observation period (2004–2023), all of which are included in the data fusion based on OMI/OMAERUV SSA products.





286 of these sites provided effective monthly SSA data during the POLDER observation period (2005–2013), which are used in the data fusion based on POLDER/GRASP SSA products. Note that all available AERONET observations are assimilated into the background satellite dataset, but only those sites with more than 10 samples during the period of OMI/POLDER are presented in the global maps of evaluation and validation, which will be described in Section 3 in detail.

### 2.3 The EnKF-based data synergy approach

Here we adopt the EnKF-based data synergy approach developed by Li et al. (2020) with several adjustments and improvements. The EnKF is a flexible data assimilation tool based on the Kalman Filter (KF), which estimates the state of a dynamic system by utilizing an ensemble of observation samples (Evensen, 1994). When assimilating ground-based site observations to satellite grid products, the resulting synergy field is a weighted average of the background satellite field and the ground-based observations. Greater weights are assigned to the estimates where the ground-based observations have lower uncertainties (i.e., smaller errors). To be specific, the ground observation $\mathbf{y}$ of a true state variable $\mathbf{x}$ for a given dynamic system model can be defined as:

$$\mathbf{y} = \mathbf{H}\mathbf{x} + \varepsilon \tag{1}$$

$\mathbf{H}$ is the observation operator that maps from the scattered observation sites to the satellite grid space, $\varepsilon$ is the observation error of ground-based measurements.

In the synergy, the state variable can be calculated as:

$$\mathbf{x^a} = \mathbf{x^b} + \mathbf{K}\mathbf{y} - \mathbf{H}\mathbf{x^a} \tag{2}$$

where the superscript $\mathbf{a}$ is the synergy field, and $\mathbf{b}$ is the background field. $\mathbf{K}$ represents the Kalman gain, expressed as:

$$\mathbf{K} = \mathbf{P}\mathbf{H^T}(\mathbf{H}\mathbf{P}\mathbf{H^T} + \mathbf{R})^{-1} \tag{3}$$

where $\mathbf{R}$ denotes the error covariance martix of the observations.

In our study, we assimilate monthly ground-based SSA observations into satellite grid SSA products at $1° \times 1°$ resolution. Specifically, the state variable $\mathbf{x}$ is the true SSA value in each $1° \times 1°$ grid. $\mathbf{y}$ are the SSA observations obtained from ground-based AERONET sites. The observation error consists of the measurement error and the representation error. The measurement error reflects the SSA errors of AERONET sunphotometer retrievals and is set to 0.03, as described in Section 2.2. The representativeness error indicates the variability of SSA within each $1° \times 1°$ grid. Here we adapt the method proposed by Li et al. (2016). The representation error is approximated as the standard deviation of all subgrid satellite SSA values within each $1° \times 1°$ grid, as described in Section 2.1.



In the EnKF, an ensemble is constructed to approximate the distribution of the state variable $\mathbf{x}$. Therefore, the sample covariance of the ensemble can be utilized to represent the true background covariance. Assumed to follow a multivariate normal distribution, the background error is expressed as the background covariance matrix $\mathbf{P}$ of the ensemble dataset $\mathbf{X}$ in the EnKF:

$$\mathbf{P} = \frac{1}{N-1}(\mathbf{X} - \overline{\mathbf{X}})(\mathbf{X} - \overline{\mathbf{X}})^{\mathbf{T}} \tag{4}$$

where $N$ is the number of samples in the ensemble. Here we construct two ensembles using all monthly mean SSA data available for Merged-OMI and Merged-POLDER separately, as described in Section 2.1.

Theoretically, EnKF assumes that the size of the ensemble dataset is sufficient to represent the bulk variance of $\mathbf{x}$ and that its distribution is ideally unbiased. However, spurious correlation may arise in practice due to insufficient samples or sampling biases in the ensemble (Anderson, 2001). This issue refers to erroneous relationships between locations that are physically distant and not meaningfully correlated, which can lead to incorrect influences of distant observations on the state variable $\mathbf{x}$ in the EnKF. To address this problem, we apply the covariance localization by truncating long-range correlations in the error covariance matrix beyond a predetermined distance $l$. Therefore, the covariance matrix accounts only for the impact of observations within the specified distance $l$ from each location, thereby avoiding the influence of distant observations (Hamill et al., 2001; Houtekamer and Mitchell, 2001). In this study, we follow the approach of Gaspari and Cohn (1999) and Li et al. (2020) to denote the localized covariance $\mathbf{P_{local}}$ as the product of the background covariance matrix $\mathbf{P}$ and the localization function $\rho$. For a given location, the correlation of observations from other locations decreases with increasing distance and becomes zero at a distance of $2c$. The relationship between $c$ and the truncation distance $l$ is generally set as (Lorenc, 2003):

$$c = \sqrt{\frac{10}{3l}} \tag{5}$$

Then the localization function $\rho$ is calculated as:

$$\rho = \begin{cases} -\frac{1}{4}(\frac{|z|}{c})^5 + \frac{1}{2}(\frac{|z|}{c})^4 + \frac{5}{8}(\frac{|z|}{c})^3 - \frac{5}{3}(\frac{|z|}{c})^2 + 1, & 0 \leq |z| \leq c \\ \frac{1}{12}(\frac{|z|}{c})^5 - \frac{1}{2}(\frac{|z|}{c})^4 + \frac{5}{8}(\frac{|z|}{c})^3 + \frac{5}{3}(\frac{|z|}{c})^2 - 5(\frac{|z|}{c}) + 4 - \frac{2}{3}(\frac{c}{|z|}), & c \leq |z| \leq 2c \\ 0, & 2c \leq |z| \end{cases} \tag{6}$$

where $z$ represents the Euclidean distance between two locations. Here we consider $l$ to be 3000 km as a global average optimal value after testing the distance from 1000 to 5000 km.

## 2.4 Evaluation approaches

To evaluate the performance of the merged dataset, we examine the statistical parameters including the linear regression equation (slope and intercept), the mean absolute bias (MAB), the correlation coefficient ($R$), and the root-mean-square error





(RMSE). Following Li et al. (2020), two cross-validation (CV) methods are applied to assess the effectiveness of the EnKF approach at locations where surface observations are not assimilated. These methods are the regional three-fold cross-validation (Region-3-CV) and the leave-one-out cross-validation (LOO-CV). The Region-3-CV is a regional adaptation of the traditional K-fold CV. Specifically, we select six typical regions globally (i.e., North Africa, the Sahel, Middle East, India, East Asia, Southeast Asia), each containing more than three AERONET sites. In each region, a site-based three-fold CV is performed, i.e., the sites are divided into three subsets, and two subsets are used for data fusion at each iteration, then the performance of the regional merged result is tested at the sites not involved in the fusion process. Regional CV is preferred over global CV due to the limited spatial representativeness of surface sites within a 3000 km distance, as discussed in Section 2.3. Remote sites are excluded from the regional validation, as they do not influence the merged result. For the LOO-CV, we still use all the sites selected in the Region-3-CV as the whole evaluation dataset and iterate through each site as a validation set, with the remaining sites used for data fusion. This process is repeated until all sites have been validated once, providing an assessment of the EnKF performance across the full dataset. Note that only AERONET sites with at least 20 samples during the OMI/POLDER period are used in both CV methods.

## 3  Results

In this study, we utilize the monthly mean SSA products from OMI and POLDER as the background SSA fields, respectively. AERONET SSA observations are assimilated into these satellite background fields using the EnKF approach, resulting in the Merged-OMI SSA data at 440 nm from October 2004 to December 2023 and the Merged-POLDER SSA data at 440, 670, 865, and 1020 nm from March 2005 to December 2013. We primarily focus on the data synergy over global land area as there are few AERONET stations located over the ocean. Additionally, the ensemble spread over ocean is relatively large with potentially large retrieval uncertainties under low aerosol loadings there (as shown in Figure S1 and Figure S2).

### 3.1  Analysis of a representative case

To illustrate how our EnKF method integrate ground based and satellite observations from different sources, we first show a representative case in the Sahel before presenting the global synergy results. We select the Banizoumbou site with more than 100 AERONET SSA observations and assimilate these observations into the satellite background field to examine the performance of the merged data at this site as well as nearby sites.

Figure 1 presents the comparison between the Merged-POLDER SSA and the original POLDER data at 670 nm for the Banizoumbou site. Similar results can be obtained from the Merged-OMI SSA at 440 nm, which are shown in Figure S3 and Figure S4. Although the original POLDER SSA (blue line in Figure 1a) shows reasonable agreement with AERONET SSA at Banizoumbou, POLDER/GRASP product tends to underestimate SSA in most cases, with several exceptions of overestimation in 2006 and 2010. These biases are largely corrected in the merged data (red line in Figure 1a). Figure 1b also demonstrates that the merged data exhibit higher consistency with AERONET compared to the original POLDER SSA, with the $R$ value increasing by 39% from 0.57 to 0.79, the MAB decreasing by 46% from 0.035 to 0.019, and the RMSE reducing by 43% from

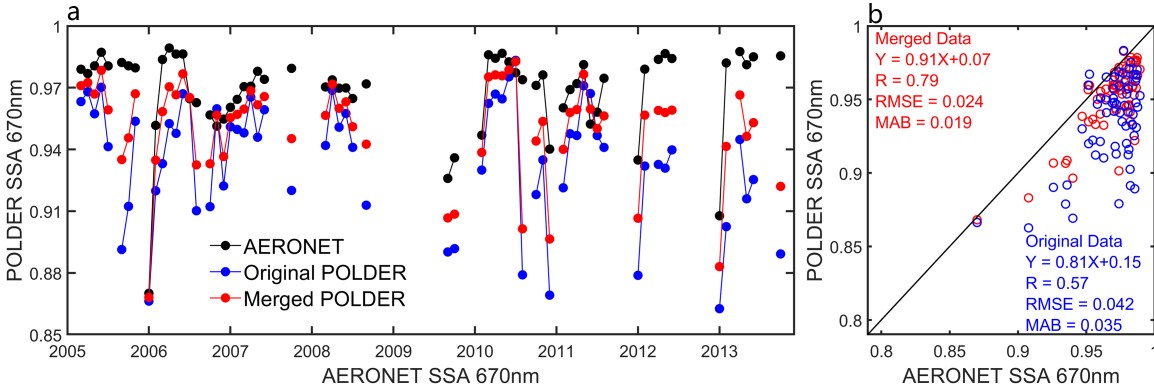

**Figure 1.** Example case of data synergy at the Banizoumbou site located in Sahel: a) the time series of monthly mean SSA at 670 nm, b) the scatter plot of the original POLDER and the merged SSA at 670 nm compared to the AERONET observations.

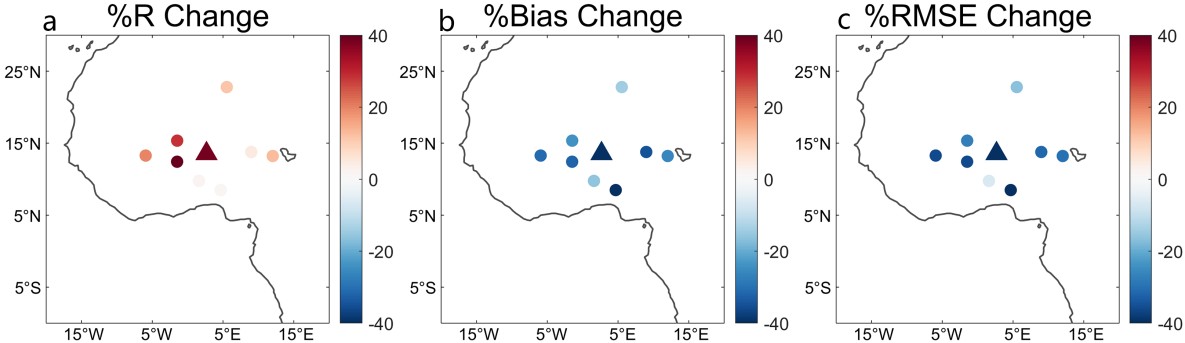

**Figure 2.** After assimilating the observations at the Banizoumbou site, the changes of a) $R$, b) MAB, and c) RMSE compared with the original POLDER SSA at 670 nm relative to AERONET sites in the Sahel.

0.042 to 0.024. These improvements are expected, as the merged results at this site represent an error weighted mean of the original POLDER retrievals and AERONET observations.

A more important concern is whether the merged data can improve SSA estimates at nearby sites, where observations are not assimilated. We examine the performance of the merged data in the entire Sahel region, and the relative changes in the accuracy

of the merged data against the original POLDER SSA are shown in Figure 2. In addition to the Banizoumbou site (marked as a triangle), SSA estimates for all the other eight nearby sites (the circles in Figure 2) show remarkable improvements, with the $R$ values increasing by 15%, the MAB decreasing by 27%, and the RMSE decreasing by 29% on average. Although the improvements at these nearby sites are less pronounced than those at Banizoumbou, the results of this case clearly demonstrate the effectiveness of the EnKF data synergy method in extending the information to a greater area. It can enhance the spatial

impact of AERONET site observations and reduce the uncertainties of satellite-derived SSA in both local and regional scales.

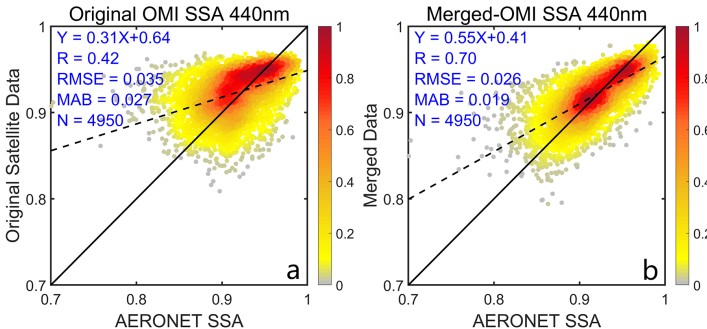

**Figure 3.** The density scatterplots of the original OMI SSA and the Merged-OMI SSA compared with AERONET SSA at 440 nm.

## 3.2 Evaluation of Merged-OMI data

The overall performance of the Merged-OMI result is examined in this section. A total of 4950 monthly SSA observations at 525 AERONET sites are assimilated into the background OMI field during the period from 2004 to 2023. Figure 3 evaluates the original OMI data and the Merged-OMI SSA at 440 nm compared with AERONET observations, respectively. The original
OMI SSA against AERONET shows a moderate correlation with an $R$ value of 0.42, an RMSE of 0.036, and a MAB value of 0.027. Many improvements are noticed in the Merged-OMI SSA, reaching a higher $R$ value of 0.70, a lower RMSE value of 0.026, and a lower MAB value of 0.019. The systematic bias is significantly corrected with the slope of the regression equation increasing from 0.31 to 0.55, which affirms the effectiveness of the EnKF approach.

We further compare the spatial distribution of the original OMI and the Merged-OMI datasets to overview the effect of
EnKF approach on a global scale. Figure 4 displays the global annual SSA distribution at 440 nm of the Merged-OMI and the original OMI data, along with their differences. There are 116 AERONET sites with more than 10 samples during the OMI period in 2004-2023, marked with circles in Figure 4. The original OMI data shows an underestimation of the annual SSA in North Africa, the Middle East, and North America; while an overestimation in the Sahel, South Africa, India, East Asia, and Southeast Asia. These biases have been largely corrected in the Merged-OMI data. Especially in the Middle East,
India, East Asia, and Southeast Asia, the changes in the annual mean SSA are ~0.02. A more detailed seasonal analysis is demonstrated in Figure 5. The seasonal performances are generally consistent with the annual results, but with some notable regional differences. Specifically, in India, East Asia, and Southeast Asia, the original OMI SSA is consistently biased high throughout the year, with the most pronounced overestimation occurring in MAM (March, April, and May). Frequent dust storms and long-range transport in MAM typically result in lower SSA in India and East Asia (Proestakis et al., 2018; Zhu
et al., 2007), whereas active biomass burning events also contribute to the lower SSA in Southeast Asia (Li et al., 2020). Consequently, the original OMI data in MAM tends to show more significant overestimation in these regions, where the largest improvements in the Merged-OMI data are also noticed in MAM. The Middle East is characterized by predominant dust aerosols and significant underestimation of the original OMI SSA is observed during active dust periods of MAM and

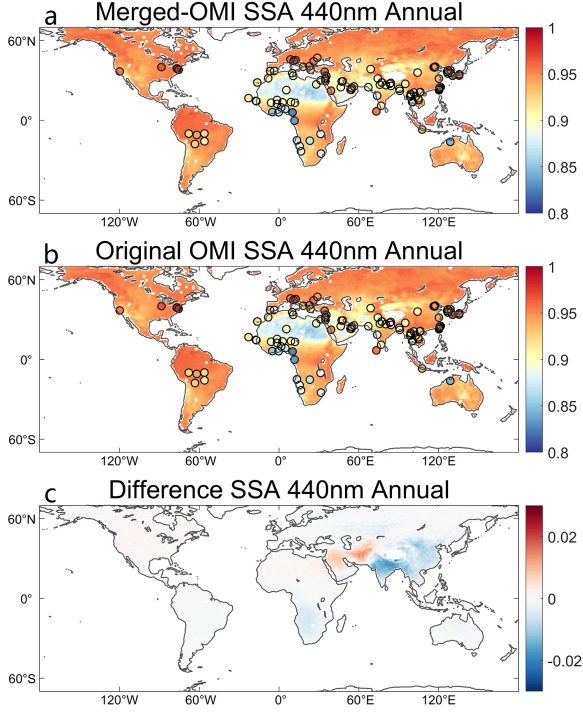

**Figure 4.** a) The annual mean SSA at 440 nm for the Merged-OMI data, b) the annual mean SSA at 440 nm for the original OMI data, and c) their differences (merged – original). Circles in a) and b) mark the observations at AERONET sites.

JJA (June, July, and August) (Engelstaedter et al., 2006), with effective corrections in the Merged-OMI data. North America

exhibits slight positive change in the Merged-OMI data to correct the underestimation of the original OMI SSA in JJA and SON (September, October, and November). In South Africa, the Merged-OMI SSA decreases the positive biases caused by the overestimation in the original OMI SSA during the biomass burning seasons in MAM, JJA, and SON. In North Africa, the original OMI SSA is overestimated in MAM and DJF (December, January, and February) but underestimated in JJA. The Merged-OMI data largely adjusts these seasonal biases, but shows an indistinct difference in the annual mean SSA due to

the balancing of seasonal over- and underestimations. Similar situations can be also observed in the Sahel, where the original SSA is significantly underestimated for dust aerosols in MAM but overestimated for biomass burning aerosols in DJF. The opposing effects of the seasonal corrections are neutralized in the Merged-OMI data, resulting in limited difference in the annual mean SSA. It is important to emphasize that the observed changes are not confined to the AERONET sites but extend to the regional scale. These results confirm that the EnKF-based fusion approach enhances the representativeness of ground-based

observations, allowing localized data to inform and improve satellite retrievals over larger regions.

To quantify the performance of the Merged-OMI dataset in different regions, we further examine the spatial distribution of the three statistical parameters (i.e., $R$, MAB, and RMSE) of the original OMI SSA compared to AERONET observations and





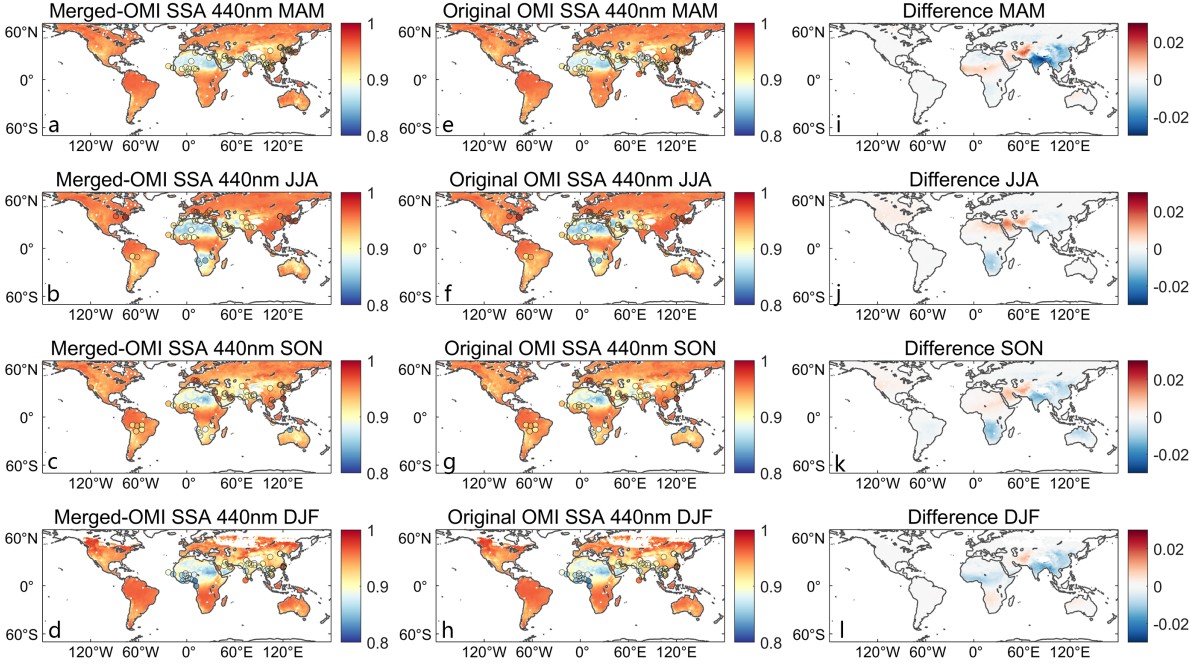

**Figure 5.** The same figure as Figure 4, but for the seasonal mean SSA at 440 nm.

their relative changes in the Merged-OMI data. Note that the $R$ values are only calculated at 55 AERONET sites with at least 20 monthly samples. As shown in Figure 6, the Merged-OMI data have substantial improvements at most AERONET sites with increases in $R$ values and reductions in both MAB and RMSE values. Especially in North Africa, the Sahel, Middle East, and India, the increase of $R$ values can reach as much as 100%, while MAB and the RMSE values decrease by ~30% or more. Moderate improvements can be observed in East and Southeast Asia. In spite of very few AERONET sites, the Merged-OMI SSA in North America and Oceania still exhibits notable reductions in MAB and RMSE values, correcting the large deviations in the original OMI data from AERONET. In South America, the original OMI SSA performs well with relatively low MAB and RMSE, and the Merged-OMI data also exhibit a slight improvement. However, the performance of Merged-OMI SSA decreased at some sites in Europe and South America. This may be attributed to insufficient spatial sampling of AERONET or large regional variabilities of SSA, which result in the poor representativeness of AERONET sites in these regions. To address this issue, our future work will further evaluate the representativeness of AERONET sites in different region to refine the EnKF data synergy method.

## 3.3 Cross validation of Merged-OMI data

To fully evaluate the performance of the data synergy, it is more critical to examine the performance at regions where ground-based observations are not assimilated or not available. For this purpose, we employ the Region-3-CV and LOO-CV as de-

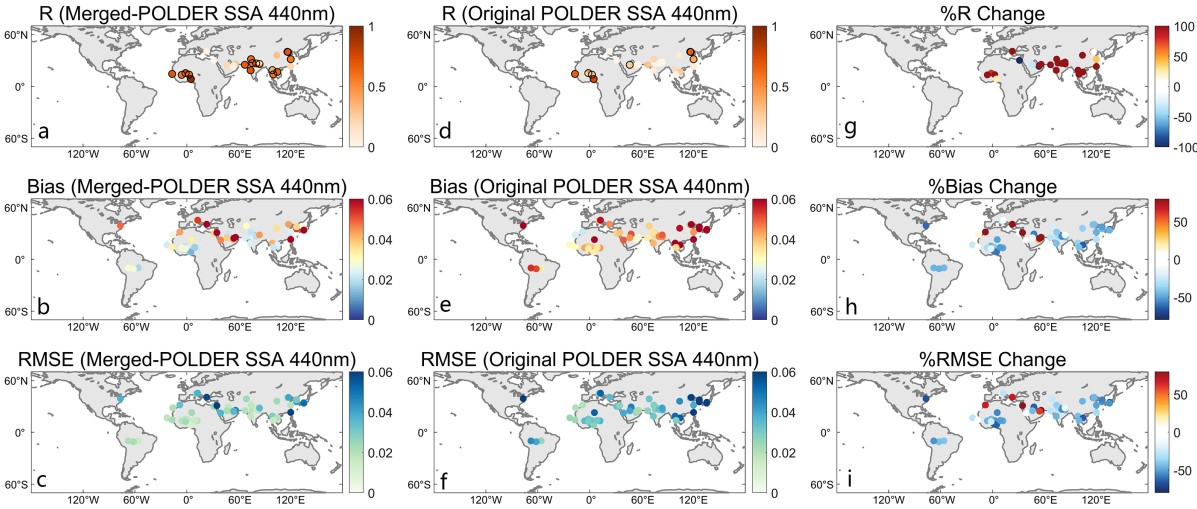

**Figure 6.** Global distribution of $R$ (top row), MAB (middle row), and RMSE (bottom row) in comparison to AERONET for the monthly mean Merged-OMI SSA at 440 nm (left column), original OMI SSA at 440 nm (middle column), and their relative changes (right column). Circled marks in a) and b) indicate significance at the 95% confidence level.

scribed in Section 2. The globe is divided into 13 regions, similar to Li et al. (2020). However, only six of them have at least three effective sites (with more than 20 samples as described in Section 2.1) during the OMI period in 2004-2023, which are then used for the validation of the results (Figure S5). These six regions are North Africa, the Sahel, Middle East, India, East Asia, and Southeast Asia, respectively.

Figure 7 evaluates the LOO-CV and the Region-3-CV OMI SSA at 440 nm against AERONET observations, respectively. Both CV schemes show notable improvements compared with the original OMI SSA. For LOO-CV results, the $R$ value increases from 0.37 to 0.47, the MAB reduces from 0.027 to 0.024, and the RMSE decreases from 0.035 to 0.032. Similar improvements are observed with the Region-3-CV, where the $R$ value increases from 0.39 to 0.52, the MAB decreases from 0.026 to 0.022, and the RMSE reduces from 0.034 to 0.030. On average, both CV methods show approximately 30% improvement in correlation, 11% and 17% in RMSE and bias reductions. These CV results confirm the effectiveness of the ENKF approach in regions where the ground observations are not assimilated.

Further spatial analysis in Figure 8 indicates that both CV schemes produce similar spatial patterns, with Region-3-CV yielding slightly better results. The majority of AERONET sites are characterized with increased $R$ values and decreased MAB and RMSE values, although the magnitude of these changes is smaller than that shown in Figure 6. Globally, Region-3-CV results show the average reductions in MAB and RMSE of 15% and 14%, respectively, with an average 70% increase in $R$ values. The decreases in MAB and RMSE can even exceed 30% at certain sites. For all the 52 validation sites, 48 sites exhibit decreased MAB and RMSE, and 46 sites show increased correlations. LOO-CV results show similar spatial pattern but weaker improvements overall. This can be attributed to the shorter distances among the sites within a region, allowing





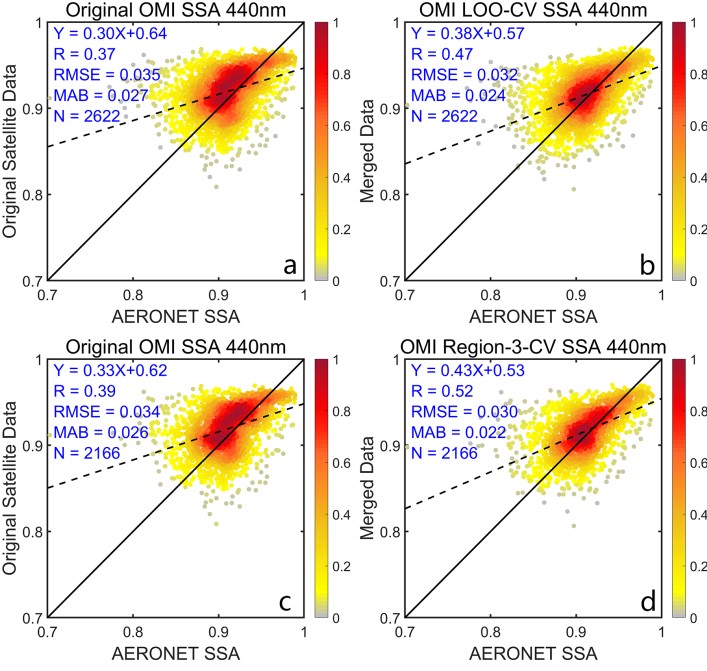

**Figure 7.** The density scatterplots of a) the original OMI SSA for LOO-CV, b) the OMI LOO-CV SSA, c) the original OMI SSA for Region-3-CV, and d) the OMI Region-3-CV SSA compared with AERONET SSA at 440 nm.

for stronger spatial correlation and influence from nearby sites. As a result, Region-3-CV can better preserve the impact of observations at closely situated AERONET sites. Note that the truncation distance we adapt in the localization function as described in Section 2.3 is an optimal value for a global scale, whereas the optimal truncation distance may vary with location. In cases where the truncation distances for some sites are smaller than our predefined threshold, LOO-CV may bring

uncertainties from the observations at remote sites with poorer representativeness and spurious correlation, which may cause weaker results than Region-3-CV. In the future, we will further investigate the differences of the representativeness and the truncation distances for ground-based sites in different regions to refine our data synergy scheme. Additionally, the lack of sufficient high-quality AERONET SSA observations makes it difficult to validate the performance of the assimilation in North America, South America, South Africa, and Oceania.

Nevertheless, both the Merged-OMI data and corresponding CV results exhibit significant improvements compared to the original OMI SSA, which indicate the effectiveness of the EnKF data synergy approach in reducing the uncertainties of background satellite SSA estimates.

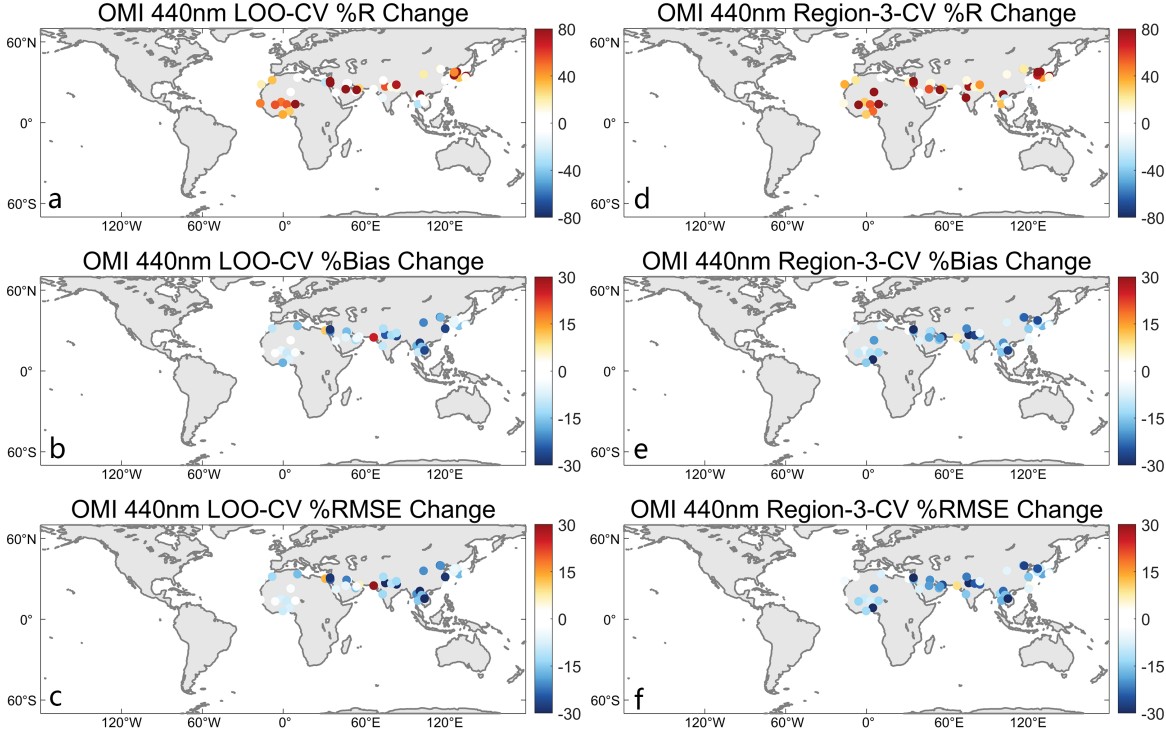

**Figure 8.** Global distribution of the overall changes in $R$ (top row), MAB (middle row), and RMSE (bottom row) for LOO-CV (left column) and Region-3-CV (right column) based on OMI SSA at 440 nm.

## 3.4 Evaluation of Merged-POLDER data

Since OMI can only provide SSA data at wavelengths up to 500 nm, a Merged-POLDER dataset is constructed to provide

SSA at visible and near-infrared bands. Specifically, we assimilate a total of ~1,800 monthly mean SSA observations at 286 AERONET sites into the background POLDER SSA field at 440, 670, 865, and 1020 nm.

Figure 9 and Figure S6 present the density scatterplots of the original POLDER data and the Merged-POLDER data with respect to AERONET SSA at four spectral bands. The original POLDER SSA shows the worst performance at 440 nm among the four wavelengths, and improves as the wavelength increases. The results for 865 nm and 1020 nm are similar to those at

670 nm. Therefore, we mainly focus on the results for 440 and 670 nm. The original POLDER SSA almost has no correlation with AERONET at 440 nm, and exhibits moderate correlations at the other three wavelengths. Significant improvements can be observed in the Merged-POLDER data at all wavelengths. The Merged-POLDER SSA at 440 nm shows an $R$ value increasing from -0.03 (not significant) to 0.47 against AERONET, with the MAB and RMSE decreasing from 0.046 and 0.060 to 0.028 and 0.041, respectively. Although the Merged-POLDER data at 440 nm shows significant improvement over the original POLDER

data, it still underperforms compared to the Merged-OMI SSA at 440 nm. Thus, we recommend using the Merged-OMI data

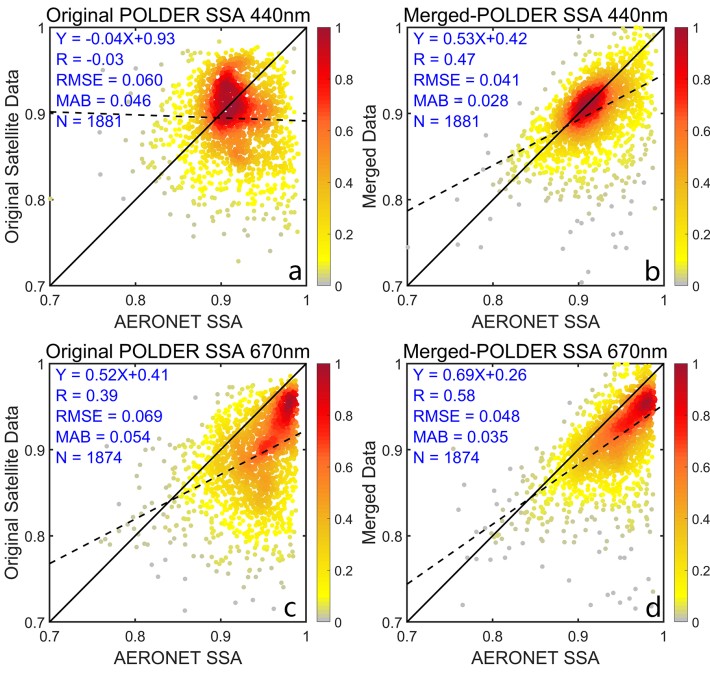

**Figure 9.** The same figure as Figure 3, but for the original and the merged SSA based on POLDER at 440 and 670 nm.

at 440 nm. At 670 nm, the $R$ value for POLDER-SSA increases from 0.39 to 0.58, and MAB and RMSE decrease from 0.054 and 0.069 to 0.035 and 0.048, respectively. Meanwhile, the original POLDER product exhibits significant systematic negative biases at 670 nm and two longer wavelengths, which are effectively corrected in the merged dataset.

Figure 10 and Figure S7 describe the spatial distribution of the annual mean SSA from the original POLDER data and
the Merged-POLDER data, alongside the observation values from 51 AERONET sites with at least 10 samples during the POLDER observation period. At 440 nm, the original POLDER data tends to overestimate the SSA for dust aerosols with a positive bias in dust source regions such as North Africa, while underestimate in other regions with a strong negative bias. These biases are notably corrected in the merged data. As for 670 nm, the Merged-POLDER data largely corrects the serious negative biases in the original POLDER data. The seasonal results generally share the same features as the annual results in
Figure 11, Figure 12, Figure S8, and Figure S9. The Merged-POLDER SSA at 440 nm evidently corrects the positive bias in dust-dominated regions, with the most notable improvement occurring in the Middle East in JJA. In other regions, the negative biases reduced in the merged data at 440 nm. Overestimations in the original data at 670 nm are mainly observed in regions with seasonal biomass burning events, such as the Middle East in JJA and SON, and the Sahel in DJF, whereas POLDER SSA tends to underestimate in the other regions. These biases are mostly reduced in the Merged-POLDER SSA. However, the
positive bias in the original data is overcorrected in the Middle East, resulting in a larger negative bias in the Merged-POLDER data. Despite this, the global performance of the Merged-POLDER data demonstrates the overall effectiveness of the EnKF data synergy technique.



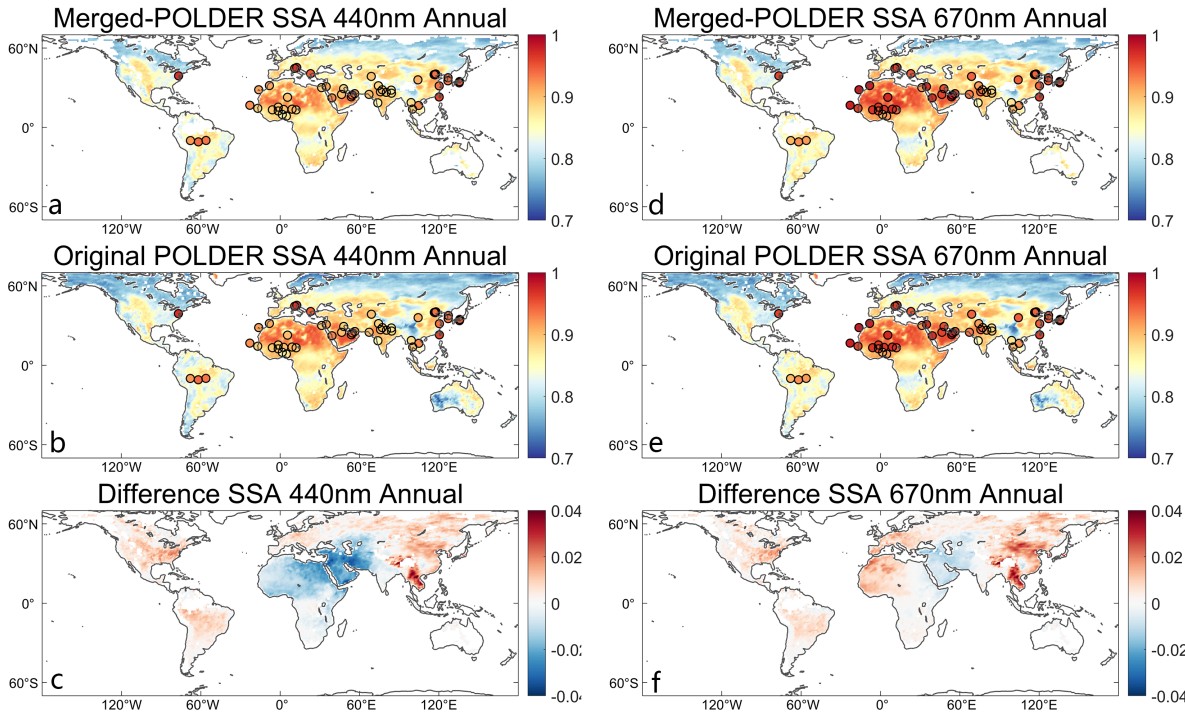

**Figure 10.** The same figure as Figure 4, but for the original and the merged SSA based on POLDER at 440 and 670 nm.

We further evaluate the performance of the Merged-POLDER dataset across different regions in Figure 13, Figure 14, Figure S10, and Figure S11. Only 25 sites provided at least 20 observation samples during the POLDER observation period, which

are used to calculate the evaluation statistics. At most of sites, the Merged-POLDER data shows a significant improvement in correlation with AERONET SSA, with substantial reductions in MAB and RMSE values. The improvement at 670 nm is similar to that at 440 nm. Notably, in Sahel, India, and Southeast Asia, the $R$ values of the Merged-PODLER data improve by up to 100% compared to the original POLDER data, with MAB and RMSE values decreasing by at least 30%. The Merged-POLDER SSA also exhibits moderate improvements in East Asia. Although there is a lack of observations in North and South

America, a significant reduction in bias and RMSE can still be noticed in the Merged-POLDER data across the two regions. However, over North Africa and the Middle East, the performance of Merged-POLDER appears less satisfactory at several sites, including Cairo_EMA_2, Dhadnah, Mezaira, Mussafa, Saada, SEDE_BOKER, Solar_Village, and Thessaloniki. Further analysis reveals that observations at these sites may have low spatial representativeness and disagree with nearby sites (see Figures S12–S19). Possible explanations are: (1) relatively high spatial variability of SSA in the surrounding regions, and

(2) data quality issues within the ensemble samples at surrounding grid boxes, such as significant negative correlations with AERONET observations and non-Gaussian error distributions (as indicated by the failure to pass the significance test in Figure S2). These factors may prevent the ensemble samples from accurately reflecting the true background covariance.

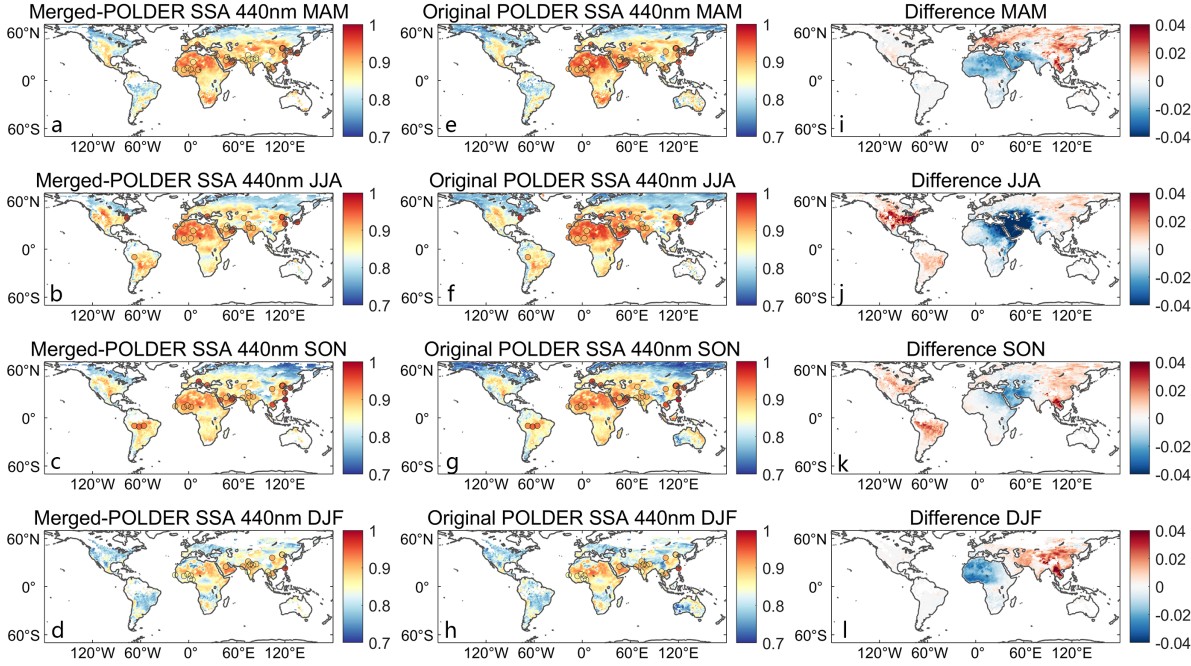

**Figure 11.** The same figure as Figure 5, but for the original and the merged SSA based on POLDER at 440 nm.

## 3.5 Cross validation of Merged-POLDER data

The Region-3-CV and LOO-CV are also performed for Merged-POLDER SSA. Note that there are only 24 effective sites with
more than 20 samples in 2005-2013 used for POLDER CV (Figure S20). Figure 15 and Figure S21 present the evaluation of
the LOO-CV and the Region-3-CV POLDER SSA with respect to AERONET observations. Similar to the OMI results, both
CV schemes applied for Merged-POLDER demonstrate significant improvements over the original POLDER SSA. At 440 nm,
LOO-CV is noticed with the $R$ value increasing from 0.37 to 0.47, the MAB reducing from 0.027 to 0.024, and the RMSE
dropping from 0.035 to 0.032. The Region-3-CV yields more pronounced improvements than the LOO-CV, where the $R$ value
increases from 0.30 to 0.49, the MAB decreases from 0.035 to 0.025, and the RMSE reduces from 0.044 to 0.034. The CV
performances for POLDER SSA at other wavelengths are similar to those at 440 nm and are provided in the supplementary
figures. These CV results further confirm the spatial extension effect of the EnKF method.

Since too few validation samples in the Region-3-CV limit the site-level evaluation, only the spatial distribution of the LOO-
CV results are presented in Figure 16 and Figure S22 . With only 24 sites available for LOO-CV in 2005-2013, the results
based on POLDER are not as satisfactory as those based on OMI. Nonetheless, the MAB and RMSE at most sites decrease in
regions except for the Middle East. For the latter, both the Merged-POLDER data and LOO-CV show decreased performance
compared to the original POLDER data, which could be possibly caused by the high aerosol variabilities and the data quality
issue of the ensemble dataset as discussed in Section 3.4.

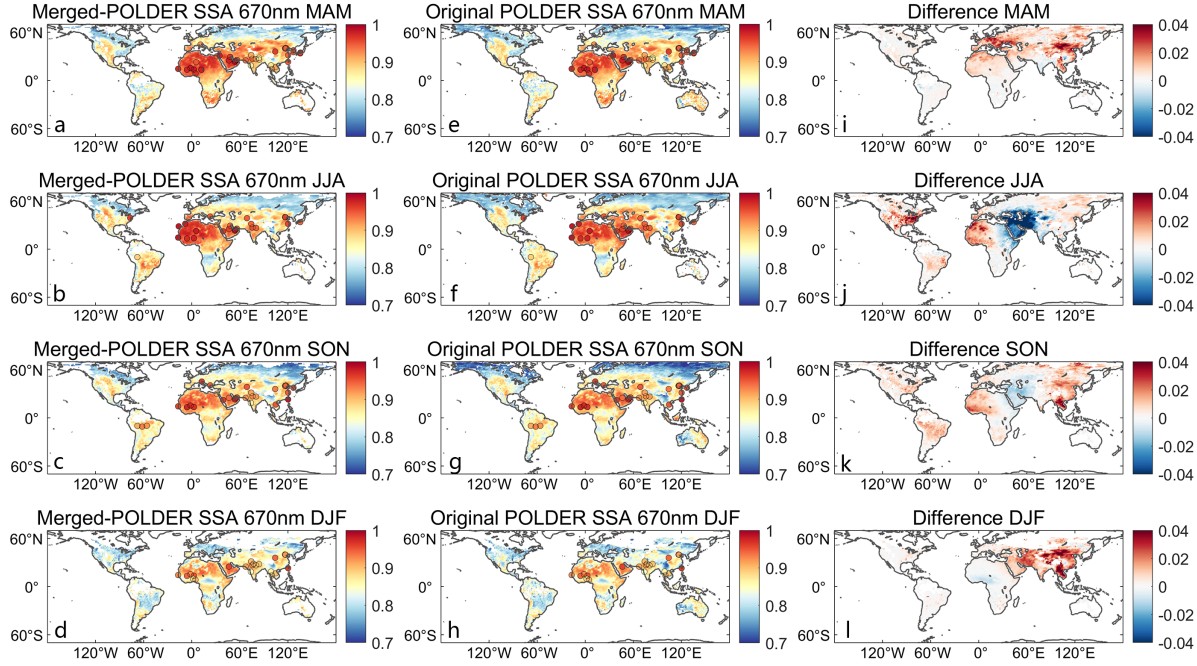

**Figure 12.** The same figure as Figure 5, but for the original and the merged SSA based on POLDER at 670 nm.

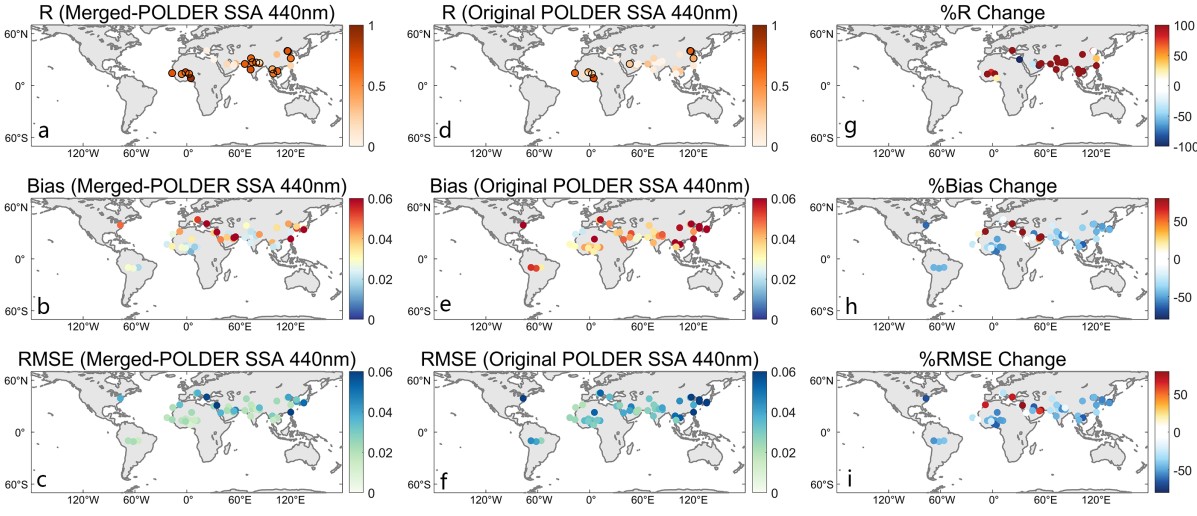

**Figure 13.** The same figure as Figure 6, but for the original and the merged SSA based on POLDER at 440 nm.

Overall, the performance of Merged-POLDER data is less satisfactory than that of Merged-OMI data, mainly due to the
shorter observation period of POLDER. On one hand, a nine-year PODLER satellite dataset result in a smaller ensemble





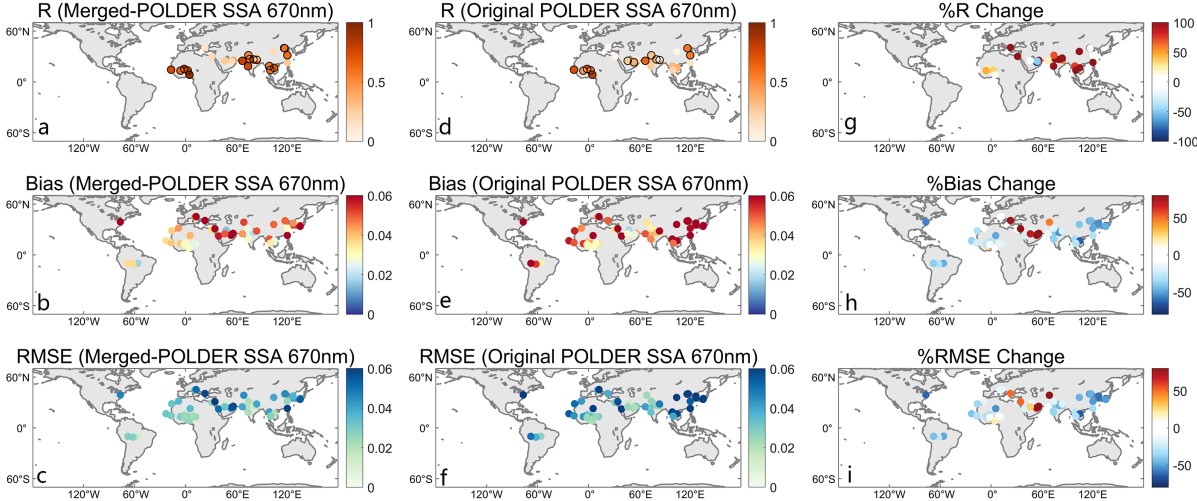

**Figure 14.** The same figure as Figure 6, but for the original and the merged SSA based on POLDER at 670 nm.

size, which introduces greater uncertainties in estimating the covariance matrix of the background field. On the other hand, POLDER SSA shows larger variability than OMI SSA at most locations (Figure S23), which adversely impact the accuracy of the synergy results. Nevertheless, the Merged-POLDER data still show significant improvements over the original datasets at all wavelengths, although the EnKF approach still requires further regional adaptation especially for regions like the Middle
East.

## 4 Discussion

As mentioned above, current SSA observations still face great challenges. In this study, the EnKF approach is utilized to derive more accurate global merged SSA data by combining the advantages of both satellite and ground-based observations. Nonetheless, there are several issues to keep in mind in the implementation of the EnKF.

The key point of EnKF is determining the errors associated with both the surface observations and the background satellite field. For surface observation errors, we represent them as the sum of measurement errors (AERONET SSA errors) and representativeness errors. The representativeness errors are estimated using the standard deviation of SSA from all $0.25°$ OMI data (or $0.1°$ POLDER data) within each $1° \times 1°$ grid box. However, even $0.1°$ grid satellite data may smooth out many of the local-scale aerosol variations, such as the differences in emission sources or topography, which could introduce uncertainties
into the merged data.

Regarding the background field errors, the ideal solution would be to construct a sufficiently large ensemble of multiple satellite products to approximate the variability of the background field. For instance, Li et al. (2020) constructed an ensemble of 474 members by combining monthly AOD data from 11 satellite datasets, which effectively captured the AOD variability



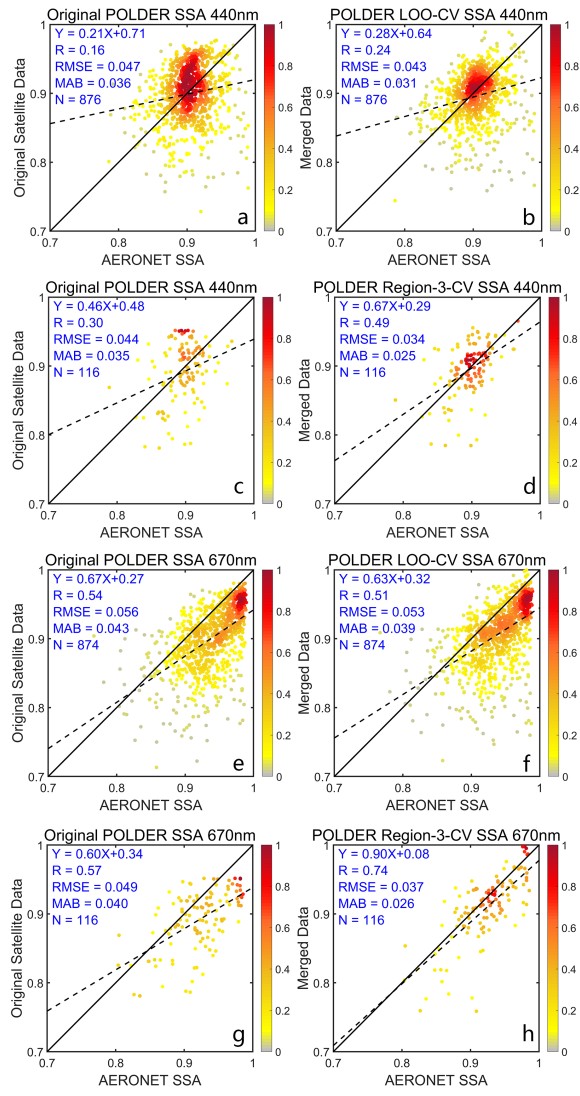

**Figure 15.** The same figure as Figure 7, but for POLDER SSA at 440 and 670 nm.

as well as its spatial covariability. However, because long-term SSA satellite products are scarce due to either insufficient

information content or high uncertainty of the observations, only POLDER and OMI SSA products are separately used to construct the background ensemble. The OMI-based ensemble and the POLDER-based ensemble contain 240 and 106 samples, respectively. Although both POLDER and OMI provide SSA retrievals at 440 nm, the differences between the two datasets are substantial due to the poor retrieval capability of POLDER at 440 nm. Thus, it is not appropriate to combine the two datasets into a single ensemble dataset.

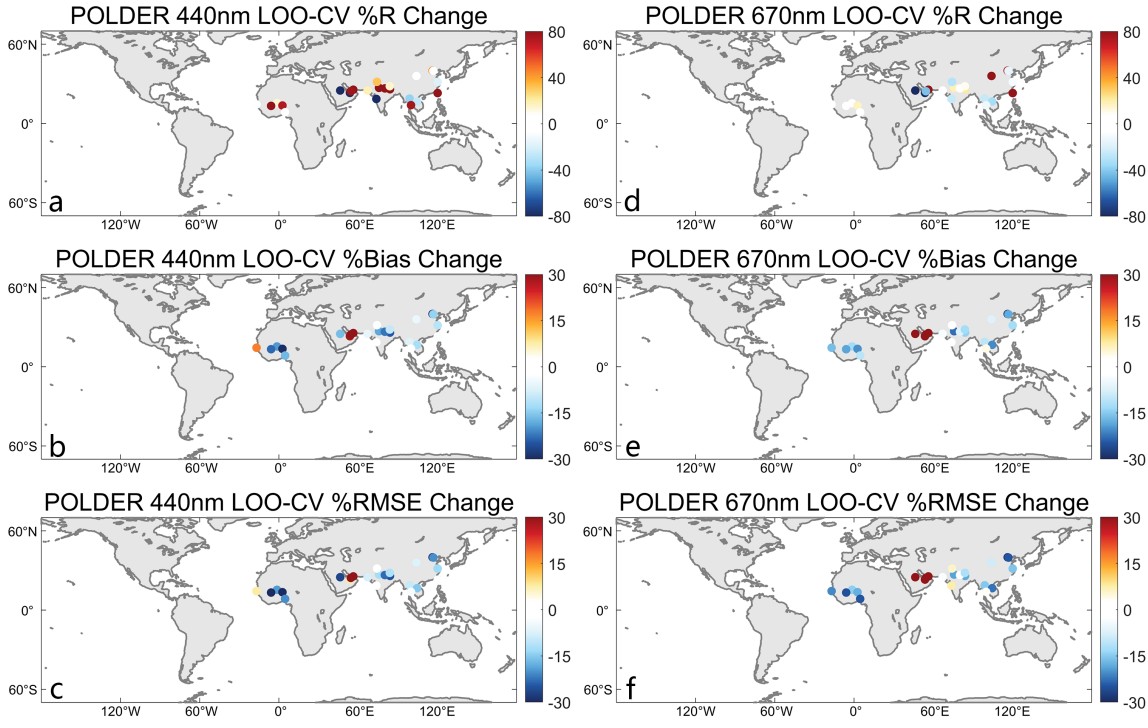

**Figure 16.** The same figure as Figure 8, but for POLDER SSA at 440 and 670 nm.

Figure 17 and Figure 18 show the comparison between the original OMI and POLDER SSA at 440 nm from 2005 to 2013. The two datasets do not exhibit a significantly positive correlation. POLDER typically overestimates SSA for dust aerosols and underestimates SSA for other aerosol types. These differences are especially pronounced in dust source regions such as North Africa and the Middle East. Although the two merged datasets significantly reduce the systemic biases in the original datasets, a large difference remains between the Merged-POLDER and Merged-OMI datasets. Further efforts are still needed to improve the accuracy of the satellite products and reduce discrepancies between datasets.

Additionally, the effectiveness of EnKF synergy method depends on the high quality and spatial representativeness of ground-based site observations. In this study, we used AERONET Level 2.0 SSA data to ensure the accuracy of SSA measurements. However, the strict quality control in Level 2.0 SSA products limits the number of available AERONET stations. Compared to AOD, Level 2.0 SSA data is provided at much fewer AERONET sites with lower temporal frequency, mainly due to the AOD threshold required in the retrieval. To ensure an SSA uncertainty within ±0.03, Level 2.0 SSA data is only available when AOD is larger than 0.4 at 440 nm, which strongly limits the number of AERONET observations available for data synergy. This also highlight the urgent need to establish a denser and higher quality surface aerosol observation network.



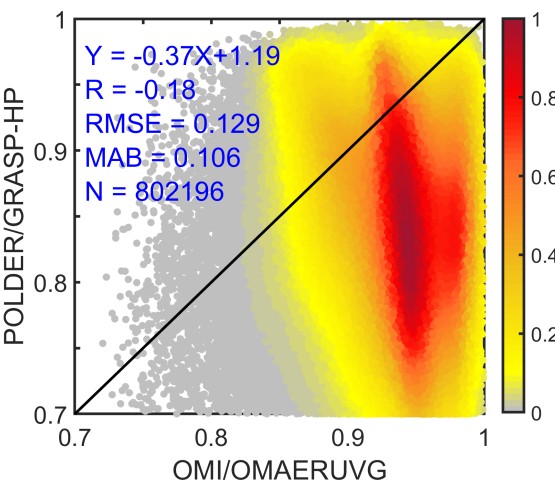

**Figure 17.** The density scatterplot of the comparison between the original OMI and POLDER SSA at 440 nm.

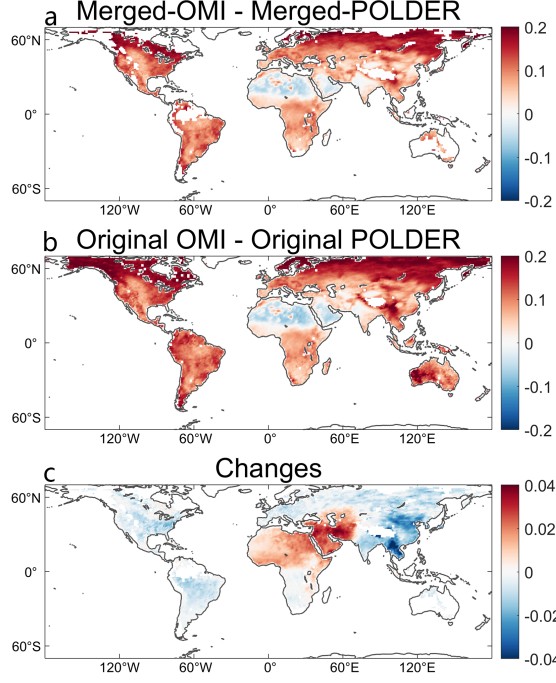

**Figure 18.** a) The differences between the Merged-OMI and the Merged-POLDER SSA at 440 nm, b) the differences between the original OMI and POLDER SSA at 440 nm, and c) their changes between a) and b).





The location of the AERONET sites is also critical as they should be preferably established in places with high repetitiveness. This is an important issue to be explored in our next study.

## 5 Data availability

The merged global land SSA datasets generated in this study are available at https://doi.org/10.5281/zenodo.14294463 (Dong, 2024).

## 6 Conclusion

In this study, we construct two high-accuracy global land SSA datasets using an EnKF data synergy approach from a combination of AERONET SSA observations and two satellite SSA datasets (i.e., POLDER and OMI). Specifically, each satellite SSA dataset is used to build an ensemble which reflects the variability of the background field. Then we assimilate AERONET SSA observations into the background field by EnKF, which can effectively propagate the effect of the individual site observations across a broader spatial extent.

Both of the Merged-OMI and the Merged-POLDER dataset demonstrate significantly higher consistency with AERONET data compared to the original satellite datasets. The global averaged $R$ values increase by up to 100%, with reductions in MAB and RMSE values more than 30%. The improvements in the two merged datasets are not constrained locally but spread in a larger region scale. Region-3-CV and LOO-CV schemes further prove the effectiveness of our method in improving the estimation at places without ground sites. For the OMI-based synergy, CV results show $R$ value increases by 70%, MAB decreases by 15%, and RMSE decreases by 14%. POLDER-based CV results also demonstrate better performance for the merged dataset although they are weaker than OMI-based CV results because of the much fewer samples in the POLDER validation dataset.

Overall, we have successfully integrated the satellite- and ground-based of SSA data using an EnKF method, despite the limited availability of both satellite products and ground-based observations. The resulting merged datasets provide more accurate SSA estimates on a global scale compared to the original satellite data, particularly in regions where ground-based observations are not available. Our merged datasets hold great potential for improving climate models and advancing our understanding of aerosol radiative effects. With the ongoing deployment of ground-based sites and advanced spaceborne sensors to monitor global aerosols, we expect to incorporate SSA data from multi-sources and generate longer-term global SSA datasets with higher accuracy.

*Author contributions.* JL designed the research. YD implemented the research. ZZ, CZ, and QL helped collect and preprocess the data. YD wrote the manuscript with contributions from all of the co-authors.



*Competing interests.* The authors declare that they have no conflict of interest.

*Acknowledgements.* We gratefully acknowledge AERONET for supporting the sunphotometer network and NASA for providing OMI data. We also appreciate the use of POLDER data, which is based on POLDER/PARASOL Level-1 data originally provided by CNES (http://www. icare.univ-lille1.fr/), processed at the AERIS/ICARE Data and Services Center with the GRASP software (https://www.grasp-open.com), developed by Dubovik et al. (2011, 2014). This study was funded by the National Natural Science Foundation of China (NSFC Grant No. 42425503), National Key Research and Development Program of China (Grant 2023YFF0805401), and NSFC Grants 52175144 and 52375121.





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
