# Peer review of "Aerosol single scattering albedo derived by merging OMI/POLDER satellite products and AERONET ground observations"

_Earth System Science Data, 2024_

## Author Comment (AC1)

**Response to Anonymous Referee #2:**

We also thank the Referee #2 for his/her comments on our work. We have carefully considered all the comments and revised the manuscript according to these comments. Below is a point-by-point response to these comments.

Aerosol single scattering albedo (SSA) is a critical parameter for characterizing aerosol scattering/absorption properties. However, obtaining accurate long-term global SSA datasets remains challenging due to limitations and uncertainties in satellite-based measurements. This manuscript constructs two long-term monthly mean SSA datasets over land by synergizing satellite observations (from OMI and POLDER) and ground-based AERONET measurements through the Ensemble Kalman Filter (EnKF) technique. The datasets provided in this study could improve the estimation of aerosol radiative effects and contribute meaningfully to climate change assessments. I would like to recommend the publication of this manuscript as the following issues are addressed.

General comments:

Could you please provide the corresponding AOD data as a quality control reference in your dataset? As you are aware, the uncertainty in the merged SSA product is closely related to AOD. Providing AOD data can help users implement effective quality control when utilizing the SSA dataset.
Response: Thank you for your suggestion. The uncertainty of SSA is indeed closely related to AOD, as also pointed out by Referee #1. In response to this issue, we have adopted an AOD threshold filtering of satellite-derived SSA, following the recommendations of Referee #1. Specifically, for the POLDER product, we retain data based on the GRASP AOD threshold of approximately 0.3 at 440 nm. For the OMI product, we follow the AERONET standard and only include data when the AOD at 440 nm exceeded 0.4. Since the resulting monthly averaged AOD values represent filtered AOD (based on quality control), rather than the actual monthly average at each grid point, we decided not to include the AOD data in the supplementary files to avoid potential misinterpretation.

Did you derive the monthly mean SSA from the original daily satellite products? If so, how was the monthly average calculated, and were any quality control measures applied during the process? These details should be clearly described in the data section.
Response: We have added the information in Section 2.1.
*"For the data fusion, the OMI SSA retrievals are interpolated to the AERONET measured wavelength of 440 nm (Dong et al. 2023). Specifically, we first calculate the monthly mean AOD and AAOD at 440 nm from daily observations, requiring at least three valid daily observations per month. The monthly SSA is then derived from the corresponding monthly AOD and AAOD. To ensure the reliability of the SSA data, AOD threshold filtering is applied to the daily data. Only daily observations with AOD at*

*440 nm greater than 0.4 are used in the monthly averaging of AOD and AAOD, consistent with the AERONET Level 2.0 criteria."* For POLDER, *"the monthly mean SSA is calculated using the same method as for OMI. However, we follow the GRASP quality control criteria by retaining only daily observations with AOD at 443 nm greater than 0.3 for the monthly averaging."*

Would it be possible to include the merged SSA results at 550 nm based on POLDER? It is commonly used in both satellite retrievals and model applications, which would enhance the usability and comparability of the dataset.

Response: Thank you for your suggestion. We have added the merged data at 550 nm based on POLDER data at 565 nm in the supplementary material (Figures S2, S6~S8, S11, S29, S30).

Why is the spatial resolution of the merged dataset set to 1°? The SSA background field can vary significantly within a 1° grid cell, which is also a key reason for the limited representativeness of ground-based observations. The dataset at a finer resolution, such as 0.5°, could potentially improve the effectiveness and applicability of ground-based observations.

Response: We set the spatial resolution to 1° primarily based on the following three considerations:

1. This resolution is appropriate for applications in climate modeling.
2. 1-degree grid cell provides a sufficient number of observations to compute representative errors at each location. This is particularly important for OMI, which has a relatively coarse spatial resolution (approximately 0.25 degrees). At 1-degree resolution, each grid cell can include up to 16 observations, allowing for more stable estimation of representative error. In contrast, a 0.5-degree grid cell would only contain about 4 observations, leading to larger random errors in the estimation.
3. The results at 1-degree and 0.5-degree resolutions are relatively similar in theory, but the 1-degree resolution significantly reduces the computational resources required for constructing the background error covariance matrix.

Specific comments:

Please specify the start and end months for 231/106 members of the ensemble in Section 2.1.

Response: We have added the information in Section 2.1.

For the global ensemble dataset, each grid cell may contain different numbers of effective members. Could it be that some grid cells have too few members? Could you provide the distribution of effective member counts across the global grid cells?

Response: We have added the information about the number of ensemble members available for each grid cell in Figure S1 and Figure S2. To ensure robust statistical representation, we required that each grid cell contain no fewer than 30 ensemble members for constructing the ensemble. This criterion has also been clarified in Section

Figure 6: The title appears to be incorrect, which should refer to the analysis based on OMI.
Response: We are sorry for the mistake. We have revised it.

Citation: https://doi.org/10.5194/essd-2024-583-RC2

---

## Author Comment (AC2)

**Response to Anonymous Referee #1:**

We thank the Referee #1 for the helpful comments on our work. We have carefully considered all the comments and revised the manuscript according to these comments. Below is a point-by-point response to these comments.

**General comments:**

This study by Dong et al. uses Ensemble Kalman Filter (EnKF) data synergy technique to construct Merged-OMI and Merged-POLDER datasets based on SSA products of AERONET, OMI and POLDER. At the same time, the results verify the effectiveness of EnKF technology in extending the information obtained from the ground station to a larger area. The constructed SSA global dataset can provide data support for global aerosol radiative forcing assessment and climate change detection. Overall, this manuscript is clearly written, and the topic is suitable for the journal. I would support for publication in the journal of Earth System Science Data after some corrections and clarifications.

Following the ESSD requirement, the author need mention the generated dataset with register doi in your abstract. And in the main text, some general description about the data structure is mandatory.

Response: Thank you for highlighting this. We have now included the registered DOI in the abstract and added a comprehensive description of the dataset's structure in Section 5, following ESSD guidelines.

General question about how you make averaging to get monthly/annual mean SSA from OMI, POLDER and AERONET? SSA is a relative value and always provided with certain AOD threshold, or its uncertainty strongly depends on the AOD levels. Therefore, a direct value average seems meaningless to me, you need average aerosol absorption and extinction, then covert to averaged SSA.

Response: We apply a consistent approach according to the reviewer's comment to calculate monthly mean SSA for OMI, POLDER, and AERONET. All the results have been updated based on the new monthly data in the manuscript. Specifically, we first compute the monthly mean AOD and AAOD, and then derived monthly SSA from these values. A daily AOD threshold filtering is also applied to ensure SSA data accuracy. For both OMI and AERONET, only daily observations with AOD at 440 nm greater than 0.4 are used in the monthly averaging of AOD and AAOD. For POLDER, we follow the GRASP algorithm threshold and include only daily data with AOD > 0.3 in the monthly SSA calculation. These details have been added to Section 2 of the revised manuscript.

**Why not merge OMI, POLDER and AERONET into one single global SSA datasets?**

Response: Although both POLDER and OMI provide SSA retrievals at 440 nm, the differences between these two datasets are substantial due to the poor retrieval

capability of POLDER at 440 nm as shown in Figure 17 and Figure 18 (as cited below). Thus, it is not appropriate to combine the two datasets into a single ensemble dataset. Nevertheless, integrating multiple satellite datasets to construct an ensemble with a sufficiently large ensemble can indeed better represent the variability of the background field. With the ongoing advancement of satellite-derived SSA products, we plan to incorporate more SSA data in future work.

Figure 17. The density scatterplot of the comparison between the two original and merged SSA datasets at 440 nm.

Figure 18. a) The differences between the Merged-OMI and the Merged-POLDER SSA at 440 nm, b) the differences between the original OMI and POLDER SSA at 440 nm, and c) their differences between a) and b).

Specific comments:

Abstract: in order to follow the ESSD requirement, you should mention the generated dataset with register doi in your abstract. Response: We have now included the DOI in the abstract.

P2 L55: I would draw your attention about two recent and studies related to your discussions by Chen et al. (2020) and Schutgens et al. (2021) about aerosol absorption evaluation and intercomparison between satellite products.

Chen, C., Dubovik, O., Schuster, G.L. et al. Multi-angular polarimetric remote sensing to pinpoint global aerosol absorption and direct radiative forcing. Nat Commun 13, 7459 (2022). https://doi.org/10.1038/s41467-022-35147-y

Schutgens, N., Dubovik, O., Hasekamp, O., Torres, O., Jethva, H., Leonard, P. J. T., Litvinov, P., Redemann, J., Shinozuka, Y., de Leeuw, G., Kinne, S., Popp, T., Schulz, M., and Stier, P.: AEROCOM and AEROSAT AAOD and SSA study – Part 1: Evaluation and intercomparison of satellite measurements, Atmos. Chem. Phys., 21, 6895–6917, https://doi.org/10.5194/acp-21-6895-2021, 2021.

Response: We greatly appreciate these references and have incorporated the discussions and citations accordingly.

"Schutgens et al. (2021) intercompared four mainstream satellite SSA products and found that SSA retrievals are generally less accurate than those of AOD. Among them, the MAP SSA product demonstrated the best performance, with the highest correlation of ~0.77 against AERONET, yet it still exhibited considerable uncertainty. Nonetheless, satellite-based SSA products remain valuable for model evaluation, as their uncertainties are significantly lower than those of current climate models. For instance, Chen et al. (2022) utilized the POLDER/GRASP product to constrain global emissions of absorbing aerosols and successfully reduced the 95% confidence interval of black carbon direct radiative forcing by a factor of two. However, this improvement remains insufficient for comprehensive climate change assessments. Considering that the uncertainty interval of ±0.03 is regarded as the upper limit for constraining aerosol radiative forcing, with errors of ±0.03 in SSA leading up to 30% uncertainties in estimating aerosol direct radiative forcing (Zhang et al., 2022), substantial efforts are still required to derive more accurate global SSA estimates."

**P3 L71: why not merge OMI, POLDER and AERONET into one single global SSA datasets?**

Response: We produce two separate merged datasets primarily because the differences between these two datasets are substantial, as illustrated in Figures 17 and Figure 18.

For the 440 nm wavelength, we recommend using the Merged-OMI dataset rather than the Merged-POLDER dataset, as discussed above.

P4 L95: could you please explicit how you do average SSA? SSA is a relative value, therefor a direct value average seems meaningless. In addition, SSA is usually provided with certain AOD threshold.

Response: We have added the information in Section 2.1 as discussed above.

"For the data fusion, the OMI SSA retrievals are interpolated to the AERONET measured wavelength of 440 nm (Dong et al. 2023). Specifically, we first calculate the monthly mean AOD and AAOD at 440 nm from daily observations, requiring at least three valid daily observations per month. The monthly SSA is then derived from the corresponding monthly AOD and AAOD. To ensure the reliability of the SSA data, AOD threshold filtering is applied to the daily data. Only daily observations with AOD at 440 nm greater than 0.4 are used in the monthly averaging of AOD and AAOD, consistent with the AERONET Level 2.0 criteria." For POLDER, "the monthly mean SSA is calculated using the same method as for OMI. However, we follow the GRASP quality control criteria by retaining only daily observations with AOD at 443 nm greater than 0.3 for the monthly averaging."

L105: could you please provide proper reference to it?

Response: We have added the relevant references in Line 123.

**Section 2.3: if I understood, is it 1D data assimilation? You correct only the grids with ground-based measurements, right?**

Response: Although we assimilate observations from discrete AERONET sites, the background field used in this study is derived from satellite-based global twodimensional SSA data. By extending the influence of these site observations, we aim to enhance SSA estimates across larger regional scales, rather than limiting improvements at the location of the surface station. Therefore, we regard this method as falling within the scope of two-dimensional data assimilation.

**Line 225: Figure 3 shows that the RMSE of SSA/original-OMI and SSA/AERONET is 0.036, while the RMSE on this figure is 0.035.**

Response: We are sorry for the mistake. We have revised it in Line 246.

Figure 4: there are less effects on Sahara Desert w/wo data assimilation, both Original and Merged-OMI seem underestimate SSA there, could you elaborate more about it? Response: On one hand, the improvement is indeed weaker over the Sahara Desert because of the large variability in the OMI background field and the limited number of ground-based observations in this region as shown in Figure S1. On the other hand, the opposing seasonal effects of the data fusion can offset each other, resulting in relatively small changes in the annual mean. As shown in Figure 5, the original OMI SSA is overestimated in MAM and DJF but underestimated in JJA in Sahara region. The Merged-OMI data largely adjusts these seasonal biases, but shows an indistinct

difference in the annual mean SSA due to the balancing of seasonal over- and underestimations.

**Figure 6: in most cases/sites, R is improved after merging, while Bias and RMSE could become worse in few sites (6h and 6i), could you explain why is that?**

Response: First, we would like to clarify that Figure 6 mistakenly presented the POLDER results instead of the OMI results. This error has been corrected in the revised version of the figure.

We now address the issue observed in Figure 6, where the Merged-POLDER dataset shows improved R but increased bias and RMSE at a few stations. Satellite retrievals consist of both systematic and random errors. In data assimilation, we expand the representativeness of ground-based observations to improve satellite SSA products. This correction is generally more effective for systematic errors. As shown in Figure 9, POLDER tends to underestimate SSA, and the merged data can substantially correct this negative bias. However, when such negative systematic bias is accompanied by random noise, some satellite retrievals may exhibit occasional overestimations of SSA. During the assimilation process, these positive deviations may be further amplified, resulting in increased bias and RMSE at certain locations. Nevertheless, such cases account for a relatively small portion of the overall dataset. Overall, the merged dataset exhibits reduced bias and RMSE compared to the original satellite products.

**L308: there is a known issue for POLDER measurements at 440 nm.**

Response: Thanks for reminding us! We have added the information in Line 330.

**Section 3.4: since you treat 4 channels (440, 670, 865, and 1020 nm) SSA for POLDER, it would be interesting you check or evaluate the SSA spectral dependence w/wo assimilation.**

Response: Thanks for this valuable question! The spectral dependence of SSA is indeed crucial for aerosol type identification. We intend to comprehensively assess the spectral dependence of SSA from both ground-based and satellite products in future research. Hence, this issue is not extensively discussed in our manuscript. However, we can present preliminary results from some typical aerosol-type stations here.

As shown in Figure R1, we examine the performance of SSA spectral dependence at four selected representative AERONET sites representing biomass burning (BB), dust, rural, and urban aerosols respectively. Generally, the merged data consistently show significant improvements across all wavelengths compared to the original satellite data, improving the representation of aerosol spectral dependence. Nevertheless, noticeable errors still remain. For example, BB aerosols show a high SSA at 440 nm, significantly decreasing with increasing wavelength based on AERONET observations. In contrast, the original satellite data presents an opposite trend for BB aerosols, with lower SSA at 440 nm that notably increases with wavelength. Although the merged data substantially correct the underestimation in the original data, it still fails to fully capture the decreasing SSA trend with increasing wavelengths. For dust aerosols, AERONET observations show low SSA at 440 nm, significantly increasing from 440 nm to 670 nm

and stabilizing thereafter. The merged data successfully address the overestimation at 440 nm and underestimation at 670 nm in the original satellite data, better representing the spectral dependence of dust aerosol SSA. Regarding rural and urban aerosols, the merged data significantly mitigate the underestimation in the original data. All of the spectral dependencies from the merged data, the original satellite data, and ground-based observations are relatively consistent.

Overall, the merged data can enhance the spectral dependence of SSA to a certain extent, though some errors persist. We plan to conduct a more systematic evaluation and analysis of SSA spectral dependencies across different data sources in future research.